# Guiding Neural Network Initialization via Marginal Likelihood Maximization

## Abstract

We propose a simple approach to help guide hyperparameter selection for neural network initialization. We leverage the relationship between neural network and Gaussian process models having corresponding activation and covariance functions to infer the hyperparameter values desirable for model initialization. Our experiment shows that marginal likelihood maximization provides recommendations that yield near-optimal prediction performance on MNIST classification task under experiment constraints. Furthermore, our empirical results indicate consistency in the proposed technique, suggesting that computation cost for the procedure could be significantly reduced with smaller training sets.

## 1 Introduction

Training deep neural networks successfully can be challenging. However, with proper initialization trained models could improve their prediction performance. Various initialization strategies in neural network have been discussed extensively in numerous research works. Glorot and Bengio (2010) focused on linear cases and proposed the normalized initialization scheme (also known as Xavier-initialization). Their derivation was obtained by considering activation variances in the forward path and the gradient variance in back-propagation. He-initialization (He et al., 2015) was developed for very deep networks with rectifier nonlinearities. Their approach imposed a condition on the weight variances to control the variation in the input magnitudes. Because of its success, He-initialization has become the de facto choice for deep ReLU networks. While Glorot- and He-initialization schemes recognize the importance of and make use of the hidden layer widths in their formulation, other methods were also suggested to improve training in deep neural networks.

Mishkin and Matas (2016) demonstrated that pre-initialization with orthonormal matrices followed by output variance normalization produces prediction performance comparable to, if not better than, standard techniques. Additionally, Schoenholz et al. (2017) developed the bound on the network depth based on the principle of 'Edge of Chaos' given a particular set of initialization hyperparameters. Furthermore, Hayou et al. (2019) showed that theoretically and in practice proper initialization parameter tuning with appropriate activation function is important to model training for improved performance.

Neal (1996) showed that as a fully-connected, single-hidden-layer feedforward untrained neural network becomes infinitely wide, Gaussian prior distributions over the network hidden-to-output weights and biases converge to a Gaussian process, under the assumption that the parameters are independent. In other words, the untrained infinite neural network and its induced Gaussian process counterpart are equivalent. Also, as a result of the central limit theorem, the covariance between network output evaluated at different inputs can be represented as a function of the hidden node activation function. Intuitively, we could therefore relate the prediction performance of an untrained, finite-width, single-hidden-layer, fully-connected feedforward neural network to a Gaussian process model with a covariance function corresponding to the network's activation function.

In this work we propose a simple and efficient method that learns from training data to guide the selection of initialization hyperparameters in neural networks. Marginal likelihood is a popular tool for choosing kernel hyperparameters in model selection. Its applications in convolutional Gaussian processes and deep kernel learning are discussed, respectively, in (van der Wilk et al., 2017; Wilson et al., 2016). Our method aims to synergize this powerful functionality of marginal likelihood and

the relationship between untrained neural networks and Gaussian process models to make recommendations for neural network initialization. We first derive the covariance function corresponding to the activation function of the network whose prediction performance we wish to evaluate. We then employ marginal likelihood optimization for the Gaussian process model to learn hyperparameters from data. We hypothesize that the optimal set of hyperparameter values could improve initialization of the neural network.

## 2 APPROACH

To assess our proposed method, we build a neural network and a Gaussian process model with corresponding activation and covariance functions. With the Gaussian process we estimate the covariance hyperparameters from training data. These hyperparameter values are then applied in the neural network to evaluate and compare its prediction accuracy among various hyperparameter sets.

We first describe the structure of the neural network, followed by the Gaussian process model and the underlying reason for employing the marginal likelihood. Then, given the network activation function we proceed to derive a closed form representation of its counterpart covariance function.

### 2.1 SINGLE-HIDDEN-LAYER NEURAL NETWORKS

Our neural network model is a fully-connected, single-hidden-layer feedforward network with 2000 hidden nodes and rectified linear unit (ReLU) activation function. Following (Lee et al., 2018), we conduct our empirical study by considering classifying MNIST images as regression prediction. Inasmuch as the network is designed for regression, we choose the mean square error (MSE) loss as its objective function, along with Adam optimizer, and accuracy as the performance metric. In addition, one-hot encoding is utilized to generate class labels, where an incorrectly labeled class is designated -0.1, and a correctly labeled class 0.9 . For example, the one-hot representation of the integer 7 is given by [-0.1, -0.1, -0.1, -0.1, -0.1, -0.1, -0.1, 0.9, -0.1, -0.1].

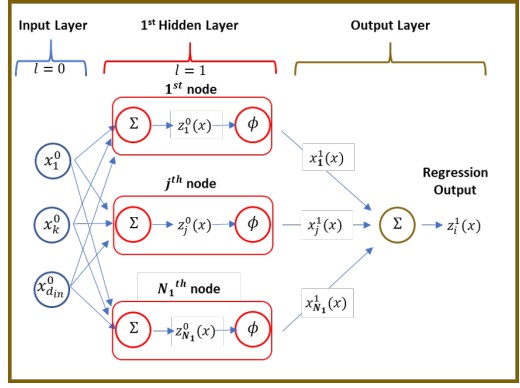 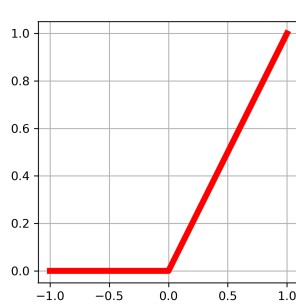

Figure 1: A single-hidden-layer, fully-connected feedforward neural network for regression prediction. Left panel: Structural diagram of the neural network. Right panel: ReLU activation function: $\phi(a) := (a)_+ = \max(0, a) = a$ for $a \geq 0$; $\phi(a) = 0$ otherwise.

As shown in the left panel of Figure (1), the single-hidden-layer neural network has a set of inputs denoted by $x = \{x_k^0\}$, $k \in \{1, 2, \cdots, d_{in}\}$ with input layer width $d_{in} = N_0 = 28\text{x}28 = 784$. The model's weight and bias parameters from $k^{th}$ input node to $j^{th}$ hidden node are $W_{jk}^0 \overset{\text{iid}}{\sim} \mathcal{N}(0, \frac{\sigma_w^2}{N_0}), b_j^0 \overset{\text{iid}}{\sim} \mathcal{N}(0, \sigma_b^2)$, and $W_{jk}^0 \perp\!\!\!\perp b_j^0$. Similarly, the weight and bias parameters from $j^{th}$ hidden node to $i^{th}$ output node with hidden layer width $N_1 = d_{in} = 2000$ are $W_{ij}^1 \overset{\text{iid}}{\sim} \mathcal{N}(0, \frac{\sigma_w^2}{N_1}), b_i^1 \overset{\text{iid}}{\sim} \mathcal{N}(0, \sigma_b^2)$, and $W_{ij}^1 \perp\!\!\!\perp b_i^1$. For regression models the output layer has a single node, and therefore $i \in \{1\}$. The ReLU nonlinearity is depicted in the right panel of Figure (1).

The input to each hidden node nonlinearity (the pre-activation) is represented by $z_j^0(x) = b_j^0 + \sum_{k=1}^{d_{in}} W_{jk}^0 x_k^0$, while the hidden unit output after the nonlinearity (the post-activation) is denoted by $x_j^1(x) = \phi(z_j^0(x))$, $j \in \{1, 2, \cdots, N_1\}$. Since we typically apply linear activation function in the output stage of a regression model, the model output is simply $z_i^1(x) = b_i^1 + \sum_{j=1}^{N_1} W_{ij}^1 x_j^1(x)$ .

## 2.2 Gaussian Processes

A Gaussian process (MacKay, 1998; Neal, 1998; Williams and Rasmussen, 2006; Bishop, 2006) is a set of random variables any finite collection of which follows a multivariate normal distribution. A Guassian process prediction model exploits this unique property and offers a Bayesian approach to solving machine learning problems. The model is completely specified by its mean function and covariance function.

By choosing a particular covariance function, a prior distribution over functions is induced which, together with observed inputs and targets, can be used to generate prediction distribution for making predictions and uncertainty measures on unknown test points. These capabilities allow Gaussian processes to be used effectively in many important machine learning applications such as human pose inference (Urtasun and Darrell, 2008) and object classification (Kapoor et al., 2010). Recent research works also apply Gaussian processes in deep structures for image classification (van der Wilk et al., 2017) and regression tasks (Wilson et al., 2016).

To help achieve optimal performance for Guassian process prediction we select a suitable covariance function and tune the model by adjusting hyperparameters characterizing the covariance function. This can be accomplished by applying the marginal likelihood which is a crucial feature that enables Gaussian processes to learn proper hyperparameter values from training data.

## 2.3 Hyperparameters and Marginal Likelihood Optimization

We briefly describe the procedure for estimating optimal hyperparamter values via maximizing the Gaussian process marginal likelihood function.

Consider a set of $N$ multidimensional input data $X = \{x_i\}_{i=1}^N$, $x_i \in \mathcal{R}^D$, and target set $y = \{y_i\}_{i=1}^N$, $y_i \in \mathcal{R}$ . For each input $x_i$ we have a corresponding input-output pair $(x_i, y_i)$, where the observed output target is given by $y_i = f(x_i) + \epsilon_i$, with data noise $\epsilon_i \sim \mathcal{N}(0, \sigma_n^2)$. We model the input-output latent function $f$ as a Gaussian process :

$$f(x_i) \sim \mathcal{GP}\big(\mu(x_i), k(x_i, x_j)\big),$$

where we customarily set the mean function $\mu(x_i) \coloneqq E[f(x_i)] = 0$, and denote $k(x_i, x_j)$ as the covariance function.

The marginal likelihood (or evidence) (Williams and Rasmussen, 2006; Bishop, 2006) measures the probability of observed targets given input data and can be expressed as the integral of the product of likelihood and the prior, marginalized over the latent function $f$:

$$p(y|X) = \int p(y, f|X)\, df = \int p(y|f, X)p(f|X)\, df. \tag{1}$$

The marginal likelihood can be obtained by either evaluating the integral (1) or by noticing $\{y_i\}_{i=1}^N = \{f(x_i) + \epsilon_i\}_{i=1}^N$, which gives us $y|X \sim \mathcal{N}(0, \mathcal{K} + \sigma_n^2 I)$ where $\mathcal{K} = [k(x_i, x_j)]_{i,j=1}^N$ and $I$ are N by N covariance matrix and identify matrix, respectively. As a result,

$$p(y|X) = \frac{1}{(2\pi)^{N/2}|\mathcal{K} + \sigma_n^2 I|^{1/2}} \exp\Big(-\frac{1}{2}y^T(\mathcal{K} + \sigma_n^2 I)^{-1}y\Big).$$

To facilitate computation, we evaluate the log marginal likelihood which is given by

$$\log p(y|X) = -\frac{1}{2}y^T(\mathcal{K} + \sigma_n^2 I)^{-1}y - \frac{1}{2}\log|\mathcal{K} + \sigma_n^2 I| - \frac{N}{2}\log 2\pi. \tag{2}$$

We are reminded here that the marginal likelihood is applied directly on the entire training dataset, rather than a validation subset. In addition, Cholesky decomposition (Neal, 1998) can be employed to calculate the term $(\mathcal{K} + \sigma_n^2 I)^{-1}$ in equation (2).

## 2.4 ReLU Covariance Function

With the structure of the single-hidden-layer ReLU neural network defined, we proceed to study its corresponding ReLU Gaussian process.

The ReLU covariance function is developed to estimate the covariance at the output of the ReLU neural network model. Our alternative derivation was inspired by the work on arc-cosine family of kernels developed in (Cho and Saul, 2009). In our work we first derive the expectation of the product of post-activations, instead of on the input to the nonlinearity (Lee et al., 2018). Then, we apply the output layer activation function on the post-activation expected value. It can be shown that the resulting representations are equivalent. The complete derivation of our expression is provided in the Appendix 5.

Referring to Figure 1, we consider input vectors $x^0, y^0 \in \mathcal{R}^{d_{in}}$. The initial weight value is drawn randomly from the Gaussian distribution $f_{W_{jk}^0} = \mathcal{N}(0, \frac{\sigma_w^2}{d_{in}})$ and bias value from $f_{b_j^0} = \mathcal{N}(0, \sigma_b^2)$. The expected value of the product of post-activations at the output of the $j^{th}$ hidden node is computed as

$$
\begin{aligned}
&E[\mathbf{X}_j(x^0)\mathbf{X}_j(y^0)] \\
&= \int \cdots \int_{-\infty}^{\infty} \max(b_j^0 + w_j^0 \cdot x^0) \max(b_j^0 + w_j^0 \cdot y^0) f_{b_j^0, W_j^0}(b, w) \, dw_j^0 \, db_j^0 \\
&= \int \cdots \int_{-\infty}^{\infty} (b_j^0 + w_j^0 \cdot x^0)_+ (b_j^0 + w_j^0 \cdot y^0)_+ f_{b_j^0, W_j^0}(b, w) \, dw_j^0 \, db
\end{aligned}
\tag{3}
$$

Suppose we denote the pre-activations as

$$
U = b_j^0 + W_j^0 \cdot x^0 = b_j^0 + \sum_{k=1}^{d_{in}} W_{jk}^0 x_k^0 \sim \mathcal{N}(0, \sigma_b^2 + \sigma_w^2 \|x\|^2),
$$

$$
V = b_j^0 + W_j^0 \cdot y^0 = b_j^0 + \sum_{k'=1}^{d_{in}} W_{jk'}^0 y_{k'}^0 \sim \mathcal{N}(0, \sigma_b^2 + \sigma_w^2 \|y\|^2).
$$

It can be shown that the random variables $U, V$ have a joint Gaussian distribution:

$$
\begin{pmatrix} U \\ V \end{pmatrix} \sim \mathcal{N}(0, \Sigma), \text{ where } \Sigma = \begin{pmatrix} \sigma_b^2 + \sigma_w^2 \|x\|^2 & \sigma_b^2 + \sigma_w^2 (x \cdot y) \\ \sigma_b^2 + \sigma_w^2 (x \cdot y) & \sigma_b^2 + \sigma_w^2 \|y\|^2 \end{pmatrix},
$$

for simplicity we let $x = x^0, y = y^0$. We can therefore write expression (3) as

$$
\iint_0^{\infty} uv \frac{1}{2\pi |\Sigma|^{\frac{1}{2}}} \exp\left( -\frac{1}{2}(u, v)\Sigma^{-1}(u, v)^T \right) du \, dv.
$$

Now let $D := |\Sigma| = \left( \sigma_b^2 + \sigma_w^2 \|x\|^2 \right)\left( \sigma_b^2 + \sigma_w^2 \|y\|^2 \right) - \left( \sigma_b^2 + \sigma_w^2 (x \cdot y) \right)^2$, and

$$
\Sigma^{-1} = \begin{pmatrix} a_{11} & a_{12} \\ a_{21} & a_{22} \end{pmatrix}, \text{ where } a_{11} = \frac{1}{D}(\sigma_b^2 + \sigma_w^2 \|y\|^2), \ a_{22} = \frac{1}{D}(\sigma_b^2 + \sigma_w^2 \|x\|^2),
$$

$$
a_{12} = a_{21} = \frac{-1}{D}(\sigma_b^2 + \sigma_w^2 (x \cdot y)).
$$

With polar coordinate transformation: $u = \frac{r}{\sqrt{a_{11}}} \cos \alpha$, $v = \frac{r}{\sqrt{a_{22}}} \sin \alpha$, expression (3) can be further reduced to

$$
\frac{1}{4\pi \mathcal{D}^{1/2} a_{11} a_{22}} \int_{\alpha=0}^{\frac{\pi}{2}} \frac{2 \sin 2\alpha}{\left( 1 - \cos\phi \sin 2\alpha \right)^2} \, d\alpha
$$

$$
= \frac{1}{2\pi \mathcal{D}^{1/2} a_{11} a_{22} \sin^3 \phi} \Big( \sin(\phi) + (\pi - \phi) \cos(\phi) \Big), \text{ where } \phi = \cos^{-1}\left( \frac{-a_{12}}{\sqrt{a_{11} a_{22}}} \right).
$$

With some algebraic operations and after computing the entries in $\Sigma^{-1}$, we arrive at

$$E[\mathbf{X}_j(x)\mathbf{X}_j(y)]$$

$$= \frac{1}{2\pi}\left(\sigma_b^2 + \|x\|^2\sigma_w^2\right)^{\frac{1}{2}}\left(\sigma_b^2 + \|y\|^2\sigma_w^2\right)^{\frac{1}{2}}\left(\sin\phi + (\pi - \phi)\cos\phi\right)$$

$$\text{where } \phi = \cos^{-1}\left\{\frac{\sigma_b^2 + (x\cdot y)\sigma_w^2}{\left(\sigma_b^2 + \|x\|^2\sigma_w^2\right)^{1/2}\left(\sigma_b^2 + \|y\|^2\sigma_w^2\right)^{1/2}}\right\}.$$

To compute the expected value, $E[\mathbf{X}_j(x)] = \int \max(b + w\cdot x)f_{b_j^0, W_{jk}^0}(b, w)\, dw\, db$, we denote $U = b + w\cdot x \sim N(0, \sigma_b^2 + \sigma_w^2\|x\|^2)$, and apply the change in variable $\frac{1}{2\sigma^2}u^2 = t$, where $\sigma^2 = \sigma_b^2 + \sigma_w^2\|x\|^2$ to obtain

$$E[\mathbf{X}_j(x)] = \int_{-\infty}^{\infty}(u)_+ f_U(u)du$$

$$= \int_0^{\infty} u\frac{1}{\sqrt{2\pi}\sigma}e^{-\frac{1}{2\sigma^2}u^2}\, du$$

$$= \int_0^{\infty} \sigma^2 dt\frac{1}{\sqrt{2\pi}\sigma}e^{-t}$$

$$= \frac{\sigma}{\sqrt{2\pi}}$$

$$= \frac{\sqrt{\sigma_b^2 + \sigma_w^2\|x\|^2}}{\sqrt{2\pi}}$$

The covariance function at the network output is therefore determined to be

$$E\left[\left(b_i^1 + \sum_{j=1}^{N_1} W_{ij}^1\mathbf{X}_j(x)\right)\left(b_i^1 + \sum_{k=1}^{N_1} W_{ik}^1\mathbf{X}_k(y)\right)\right] - E\left[b_i^1 + \sum_{j=1}^{N_1} W_{ij}^1\mathbf{X}_j(x)\right]\left[b_i^1 + \sum_{k=1}^{N_1} W_{ik}^1\mathbf{X}_k(y)\right]$$

$$= E[(b_i^1)^2] + \sum_{j=1}^{N_1} E[(W_{ij}^1)^2]E[\mathbf{X}_j(x)\mathbf{X}_j(y)] - \frac{1}{2\pi}\sqrt{\sigma_b^2 + \sigma_w^2\|x\|^2}\sqrt{\sigma_b^2 + \sigma_w^2\|y\|^2}$$

$$= \sigma_b^2 + \frac{\sigma_w^2}{N_1}N_1 E[\mathbf{X}_j(x)\mathbf{X}_j(y)] - \frac{1}{2\pi}\sqrt{\sigma_b^2 + \sigma_w^2\|x\|^2}\sqrt{\sigma_b^2 + \sigma_w^2\|y\|^2}$$

$$= \sigma_b^2 + \frac{\sigma_w^2}{2\pi}\left(\sigma_b^2 + \|x\|^2\sigma_w^2\right)^{\frac{1}{2}}\left(\sigma_b^2 + \|y\|^2\sigma_w^2\right)^{\frac{1}{2}}\left(\sin\phi + (\pi - \phi)\cos\phi - 1\right). \qquad \blacksquare$$

## 2.5 Gaussian Process Prediction: A Simulation

Performing simulations allows us to explore and understand some properties of the models we wish to study. Simulation results also offer the opportunity for evaluating model precision and insight into observed events.

To demonstrate making predictions with Gaussian process regression model, we borrow equations from (Williams and Rasmussen, 2006) where the formulation of Gaussian process predictive distribution is treated in great detail.

Given the design matrix $X = \{x_i\}_{i=1}^N$, $x_i \in \mathcal{R}^D$, observed targets $y = \{y_i\}_{i=1}^N$, $y_i \in \mathcal{R}$, unknown test data $X_*$, and their function values $f_* := f(X_*)$, the joint distribution of the target and function values is computed as

$$\begin{bmatrix} y \\ f_* \end{bmatrix} \sim N\left(0, \begin{bmatrix} K(X, X) + \sigma_n^2 I & K(X, X_*) \\ K(X_*, X) & K(X_*, X_*) \end{bmatrix}\right),$$

where $K(X, X)$ represents the covariance matrix of all pairs of training points, $K(X, X_*)$ denotes that of pairs of training and test points, and $K(X_*, X_*)$ gives the covariance matrix of pairs of test points.

The prediction distribution is the conditional distribution

$$f_* | X, y, X_* \sim N(\mu_*, \Sigma_*)$$

with mean function $\mu_* = K(X_*, X)[K(X, X) + \sigma_n^2 I]^{-1} y$

and covariance $\Sigma_* = K(X_*, X_*) - K(X_*, X)[K(X, X) + \sigma_n^2 I]^{-1} K(X, X_*).$

The simulation starts out with setting the hyperparameters of the ReLU covariance function to $(3.6, 0.02)$, chosen from $\sigma_w^2 \in [0.4, 1.2, 2.0, 2.8, 3.6]$, and $\sigma_b^2 \in [0.0001, 0.01, 0.02]$. We randomly select a set of 70 training and 30 test location points from 100 values evenly spaced in the interval $[0.0, 1.0]$. Ten sample paths, as shown in the top left panel of Figure (2), are generated from the design Gaussian process model. Their sample mean produces 70 training target and 30 test values. We then estimate the optimal hyperparameters from the training targets via evaluating the marginal likelihood, equation (2), over the design ranges of $\sigma_w^2$ and $\sigma_b^2$.

The maximum marginal likelihood is obtained at $\{\tilde{\sigma}_w^2, \tilde{\sigma}_b^2\} = \{3.6, 0.02\}$ which is the design hyperparameter pair. The minimum marginal likelihood is obtained at $\{\hat{\sigma}_w^2, \hat{\sigma}_b^2\} = \{3.6, 0.0001\}$. A Gaussian process model is then built with the optimal hyperparameter pair to make predictions for the 30 test location points. The model accuracy is assessed with a RMSE of 0.00051. Additionally we overlay the predicted and true test target values, as shown in the top middle panel of Figure (2), to detect any prediction errors. We plot the line of equality to further validate the estimated hyperparameters, as depicted in the top right panel of the figure.

The evaluation process is repeated applying the hyperparameter pair $\{\hat{\sigma}_w^2, \hat{\sigma}_b^2\}$ which produces a prediction RMSE of 0.00188, over 3 times as large as the optimal case. The accuracy plots shown in the bottom panels of Figure (2) indicate some prediction errors.

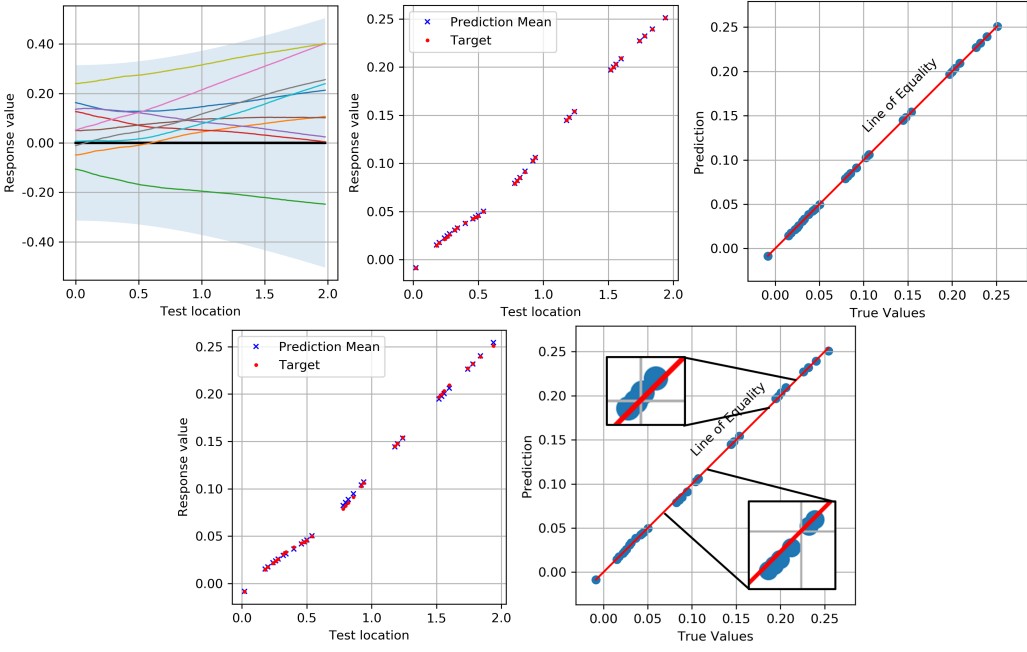

Figure 2: Gaussian process regression prediction on simulated data. Top left: 10 sample paths generated from a Gaussian process model with hyperparameters $(\sigma_w^2, \sigma_b^2) = (3.6, 0.02)$. Top middle: Point-wise visual comparison between predicted and true target values for the optimal hyperparameter pair $\{3.6, 0.02\}$, showing good prediction results. Top right: The line of equality further confirming the prediction accuracy. Bottom left: Point-wise visual comparison for hyperparameter pair $\{3.6, 0.0001\}$. Bottom right: Prediction errors revealed with the line of equality.

Our simulation results agree with the principle that through optimizing the marginal likelihood of the Gaussian process model, we could estimate from training data the hyperparameter values most appropriate for its chosen covariance function.

## 3 MNIST CLASSIFICATION EXPERIMENT

We conduct a classification experiment on the MNIST handwritten digit dataset (LeCun, 1998) making use of corresponding ReLU neural network and Gaussian process models. As in (Lee et al., 2018), the classification task on the class labels is treated as Gaussian process regression (also known as kriging in spatial statistics (Cressie, 1993)).

It is necessary to point out that the goal of this work is to examine using the marginal likelihood to estimate the best available initial hyperparameter setting for neural networks, rather than determining the networks' optimal structure.

Our experiment consists of three main steps: (A) searching within a given grid of hyperparameter values for the pair $\{\tilde{\sigma}_w^2, \tilde{\sigma}_b^2\}$ that maximizes the log marginal likelihood function of the Gaussian process model, (B) evaluating prediction accuracy of the corresponding neural network at each grid point $\{\sigma_w^2, \sigma_b^2\}$ including $\{\tilde{\sigma}_w^2, \tilde{\sigma}_b^2\}$, and (C) assessing neural network performance over all tested hyperparameter pairs.

### 3.1 PROCEDURE

The workflow for the experiment is as follows: we set up a grid map of $\sigma_w^2 \in \{0.4, 1.2, 2.0, 2.8, 3.6\}$, $\sigma_b^2 \in \{0.0, 1.0, 2.0\}$. Then, N samples are randomly selected from the MNIST training set to form a training subset, where N is the training size. This is followed by computing the log marginal likelihood (equation 2) at each grid point. This allows us to identify the hyperparameter pair $\{\tilde{\sigma}_w^2, \tilde{\sigma}_b^2\}$ that yields the maximum log marginal likelihood value.

On the neural network side, we build a fully-connected feedforward neural network with a single hidden layer width, hidden_width, of 2000 nodes, Adam optimizer, and mse loss function. Since the network model is fully connected, the size of the input layer $d_{in}$ is 28(pixels) x 28(pixels) = 784. Prior to training, the initialization parameters $\{w, b\}$ are set by sampling the distributions $\mathcal{N}(0, \sigma_w^2/d_{in})$ and $\mathcal{N}(0, \sigma_b^2)$ for weights and biases from the input to the hidden layer, and $\mathcal{N}(0, \sigma_w^2/2000)$ and $\mathcal{N}(0, \sigma_b^2)$ for weights and biases from the hidden to the output layer. The neural network is then trained with the training subset generated previously. We compute the model classification accuracy on the MNIST test set and repeat the procedure over the entire grid map of hyperparameter pairs.

To investigate the usefulness of our proposed approach for assisting model initialization, we employ He-initialization approach as a benchmark to measure numerically and graphically our neural network performance over all tested hyperparameter pairs. Additionally, we check for recommendation consistency.

### 3.2 RESULTS

Applying the method described in Section 2.3 for estimating model hyperparameter pair we obtain a consistent recommendation of $(\sigma_w^2, \sigma_b^2) = (3.6, 0.0)$.

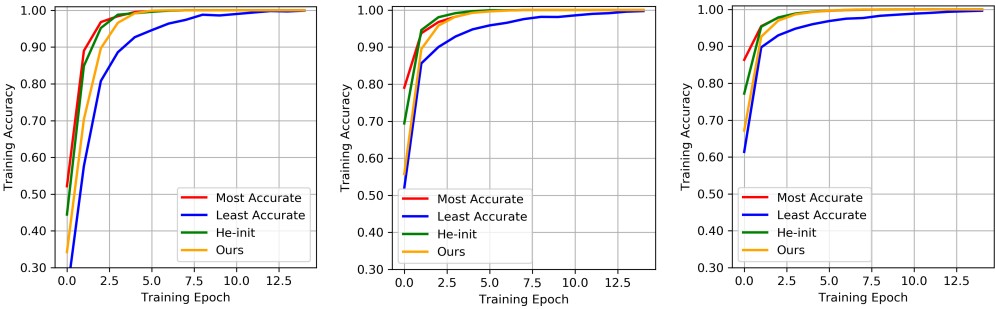

Figure 3: Comparing MNIST training accuracy over various training sizes. We observe that the convergence rate based on our method approaches that using He-initialization as the training size increases. This suggests that our technique may potentially be efficient for guiding deep neural network initialization. Left: train_size=1000. Middle: train_size=3000. Right: train_size=5000.

After running 250 training epochs, convergence of the neural network model and its prediction accuracy are studied for different training sizes. We observe that training based on our initialization approach converges to that based on He-initialization as the size of training samples increases, as shown in Figure 3. This seems to suggest that our approach may be used as an efficient tool for recommending initialization in deep learning.

It is worth noting that the Gaussian process model marginal likelihood consistently suggests the hyperparameter pair $(\sigma_w^2, \sigma_b^2) = (3.6, 0)$. The fact that the bias variance $\sigma_b^2$ is estimated to be 0 coincides with the assumption that bias vector being 0 in (He et al., 2015).

Table 1 lists neural network model prediction accuracy based on, respectively, our approach and He-initialization scheme, against the best and the worst performers. The results indicate that more frequently our approach achieves slightly better accuracy than based on He-initialization. However, neither approach reliably gives an estimate of weight variance close to that for the best case.

Table 1: Single-hidden-layer fully-connected neural network model prediction accuracy on MNIST test set, and associated hyperparameter pair.

| Size | Best Case Acc. | $(\sigma_w^2, \sigma_b^2)$ | Worst Case Acc. | $(\sigma_w^2, \sigma_b^2)$ | He-Init Acc. | $(\sigma_w^2, \sigma_b^2)$ | Ours Acc. | $(\sigma_w^2, \sigma_b^2)$ |
|---|---|---|---|---|---|---|---|---|
| 10000 | 96.85 | (2, 0) | 96.04 | (0.4, 2) | **96.85** | **(2, 0)** | 96.60 | (3.6, 0) |
| 20000 | 97.25 | (2.8, 0) | 96.70 | (3.6, 1) | 97.01 | (2, 0) | **97.09** | **(3.6, 0)** |
| 30000 | 97.50 | (1.2, 0) | 96.91 | (2, 2) | 97.07 | (2, 0) | **97.29** | **(3.6, 0)** |
| 40000 | 97.43 | (0.4, 0) | 97.16 | (0.4, 2) | 97.35 | (2, 0) | **97.42** | **(3.6, 0)** |
| 50000 | 97.71 | (3.6, 0) | 97.29 | (0.4, 2) | 97.50 | (2, 0) | **97.71** | **(3.6, 0)** |

## 4 DISCUSSION AND FUTURE WORK

In this work we propose a simple, consistent, and time-efficient method to guide the selection of initial hyperparameters for neural networks. We show that through maximizing the log marginal likelihood we can learn from training data hyperparameter setting that leads to accurate and efficient initialization in neural networks.

We develop an alternative representation of the ReLU covariance function to estimate the covariance at the output of the ReLU neural network model. We first derive the expectation of the product of post-activations. Then, we apply the output layer activation function on the post-activation expected value to generate the output covariance function. Utilizing marginal likelihood optimization with the derived ReLU covariance function we perform a simulation to demonstrate the effectiveness of Gaussian process regression.

We train a fully-connected single-hidden-layer neural network model to perform classification (treated as regression) on MNIST data set. The empirical results indicate that applying the recommended hyperparameter setting for initialization the neural network model performs well, with He-initialization scheme as the benchmark method.

A further examination of the results reveals consistency of the process. This implies that smaller training subsets could be used to provide reasonable recommendation for neural network initialization on sizable training data sets, reducing the computation time which is otherwise required for inverting considerably large covariance matrices.

The main goal of our future research is to investigate if our proposed method is adequate for deep neural networks with complicated data sets. We wish to ascertain if consistent recommendation could be attained by learning from larger data sets of color images via marginal likelihood maximization. We will attempt to derive or approximate multilayer covariance functions corresponding to various activation functions. Deep fully-connected neural network models will be built to perform classification on CIFAR-10 data set. Our hypothesis is that learning directly from training data helps to improve neural network initialization strategy.

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

## 5 APPENDIX

**Covariance Function at the Output of ReLU Neural Network**

Our derivation follows the work on arc-cosine family of kernels developed in (Cho and Saul, 2009). However, instead of applying coplanar vector rotation in calculating the kernel integral, we recognize that the integrand can be written in terms of two jointly normal random variables. This helps to facilitate the computation which becomes more involved when both the weight and bias parameters are included.

The derivation is also made to conform to the arc-cosine kernel by utilizing the identities (Cho and Saul, 2009, equation (17), (18)) to give us

$$
\int_{\eta=0}^{\frac{\pi}{2}} \frac{1}{1 - \cos\phi \, \cos\eta} \, d\eta = \frac{\pi - \phi}{\sin\phi},
$$

$$
\int_{\theta=0}^{\frac{\pi}{2}} \frac{\sin 2\theta}{\left(1 - \cos\phi \, \sin 2\theta\right)^2} \, d\theta
$$

$$
= \frac{1}{\sin^3\phi} \Big( \sin(\phi) + (\pi - \phi)\cos(\phi) \Big). \tag{4}
$$

Equation (4) is derived, with the substitution $\eta = 2(\theta - \frac{\pi}{4})$, as follow:

$$
\int_{\theta=0}^{\frac{\pi}{2}} \frac{\sin 2\theta}{\left(1 - \cos\phi \, \sin 2\theta\right)^2} \, d\theta
$$

$$
= \int_{\eta=0}^{\frac{\pi}{2}} \frac{\cos\eta}{\left(1 - \cos\phi \, \cos\eta\right)^2} \, d\eta
$$

$$
= \frac{\partial}{\partial\cos\phi} \int_{\eta=0}^{\frac{\pi}{2}} \frac{1}{1 - \cos\phi \, \cos\eta} \, d\eta
$$

$$
= \frac{\partial}{\partial\cos\phi} \Big(\frac{\pi - \phi}{\sin\phi}\Big) = \frac{-1}{\sin(\phi)} \frac{\partial}{\partial\phi} \Big(\frac{\pi - \phi}{\sin\phi}\Big)
$$

$$
= \frac{1}{\sin^3\phi} \Big( \sin(\phi) + (\pi - \phi)\cos(\phi) \Big). \qquad \blacksquare
$$

Denote the input layer (layer 0) weight and bias parameters as $b_j^0 \sim \mathcal{N}(0, \sigma_b^2)$ and $W_{jk}^0 \overset{\text{iid}}{\sim} \mathcal{N}(0, \frac{\sigma_w^2}{d_{in}})$, where $b_j^0 \perp\!\!\!\perp W_{jk}^0$ for all $k \in \{1, \cdots, d_{in}\}, j \in \{1, \cdots, N_1\}$.

The expected value of the product of post-activations at the output of the $j^{th}$ hidden node is computed as

$$
E[\mathbf{X}_j(x^0)\mathbf{X}_j(y^0)]
$$

$$
= \int \cdots \int_{-\infty}^{\infty} \max(b_j^0 + w_j^0 \cdot x^0) \max(b_j^0 + w_j^0 \cdot y^0) f_{b_j^0, W_j^0}(b, w) \, dw_j^0 \, db_j^0
$$

$$
= \int \cdots \int_{-\infty}^{\infty} (b_j^0 + w_j^0 \cdot x^0)_+ (b_j^0 + w_j^0 \cdot y^0)_+ f_{b_j^0, W_j^0}(b, w) \, dw_j^0 \, db \tag{5}
$$

Each pre-activation can be written in terms of a random variable:

$$
U = b_j^0 + W_j^0 \cdot x^0 = b_j^0 + \sum_{k=1}^{d_{in}} W_{jk}^0 x_k^0 \sim \mathcal{N}(0, \sigma_b^2 + \sigma_w^2 \|x\|^2),
$$

$$
V = b_j^0 + W_j^0 \cdot y^0 = b_j^0 + \sum_{k'=1}^{d_{in}} W_{jk'}^0 y_{k'}^0 \sim \mathcal{N}(0, \sigma_b^2 + \sigma_w^2 \|y\|^2).
$$

Since $E[U] = E[V] = 0$, their covariance can be expressed as

$$\text{cov}(U, V) = E[(b_j^0 + W_j^0 \cdot x^0)(b_j^0 + W_j^0 \cdot y^0)]$$

$$= E[(b_j^0)^2] + E\Big[\sum_{k=1}^{d_{in}} \sum_{k'=1}^{d_{in}} W_{jk}^0 W_{jk'}^0 x_k^0 y_{k'}^0\Big]$$

$$= \sigma_b^2 + \sum_{k=1}^{d_{in}} \sum_{k'=1}^{d_{in}} E\Big[W_{jk}^0 W_{jk'}^0\Big] x_k^0 y_{k'}^0$$

$$= \sigma_b^2 + \sigma_w^2 \sum_{k=1}^{d_{in}} x_k^0 y_k^0$$

$$= \sigma_b^2 + \sigma_w^2 (x \cdot y) \quad \text{(For simplicity we set } x = x^0, y = y^0)$$

This implies that the random variables $U, V$ have a joint Gaussian distribution:

$$\begin{pmatrix} U \\ V \end{pmatrix} \sim \mathcal{N}(0, \Sigma), \text{ where } \Sigma = \begin{pmatrix} \sigma_b^2 + \sigma_w^2 \|x\|^2 & \sigma_b^2 + \sigma_w^2 (x \cdot y) \\ \sigma_b^2 + \sigma_w^2 (x \cdot y) & \sigma_b^2 + \sigma_w^2 \|y\|^2 \end{pmatrix}.$$

We can, therefore, rewrite equation (5) as

$$\iint_0^\infty uv \frac{1}{2\pi |\Sigma|^{\frac{1}{2}}} \exp\big( -\frac{1}{2}(u, v)\Sigma^{-1}(u, v)^T \big) \, du \, dv \tag{6}$$

Denote $D := |\Sigma| = \big(\sigma_b^2 + \sigma_w^2 \|x\|^2\big)\big(\sigma_b^2 + \sigma_w^2 \|y\|^2\big) - \big(\sigma_b^2 + \sigma_w^2 (x \cdot y)\big)^2$, and

$$\Sigma^{-1} = \begin{pmatrix} a_{11} & a_{12} \\ a_{21} & a_{22} \end{pmatrix},$$

with $a_{11} = \dfrac{1}{D}(\sigma_b^2 + \sigma_w^2 \|y\|^2)$, $a_{22} = \dfrac{1}{D}(\sigma_b^2 + \sigma_w^2 \|x\|^2)$, and
$a_{12} = a_{21} = \dfrac{-1}{D}(\sigma_b^2 + \sigma_w^2 (x \cdot y))$.

We therefore have:

$$D(a_{11}a_{22} - a_{12}^2) = D\Big(\frac{1}{D}(\sigma_b^2 + \sigma_w^2 \|y\|^2)\frac{1}{D}(\sigma_b^2 + \sigma_w^2 \|x\|^2) - \big(\frac{-1}{D}(\sigma_b^2 + \sigma_w^2 (x \cdot y))^2\big)\Big)$$

$$= \frac{1}{D}\Big(\big(\sigma_b^2 + \sigma_w^2 \|x\|^2\big)\big(\sigma_b^2 + \sigma_w^2 \|y\|^2\big) - \big(\sigma_b^2 + \sigma_w^2 (x \cdot y)\big)^2\Big)$$

$$= 1. \quad \text{(by definition)} \tag{7}$$

The exponential term in equation (6) then becomes:

$$-\frac{1}{2}(u, v)\Sigma^{-1}(u, v)^T = -\frac{1}{2}\big(a_{11}u^2 + 2a_{12}uv + a_{22}v^2\big).$$

We now make use of the transformation from Cartesian to polar coordinates by setting

$$u = \frac{r}{\sqrt{a_{11}}} \cos \alpha, \; v = \frac{r}{\sqrt{a_{22}}} \sin \alpha$$

$$\implies a_{11}u^2 = r^2 \cos^2 \alpha, \; a_{22}v^2 = r^2 \sin^2 \alpha.$$

The Jacobian $\mathcal{J}$ is calculated as

$$\Big|\frac{\partial(u, v)}{\partial(r, \alpha)}\Big| = \frac{r}{\sqrt{a_{11}a_{22}}}.$$

Equation (6) can in turn be expressed as

$$\frac{1}{2\pi\mathcal{D}^{1/2}}\int_{\alpha=0}^{\frac{\pi}{2}}\int_{r=0}^{\infty}\frac{r^2\sin 2\alpha}{2\sqrt{a_{11}a_{22}}}\exp\Big(\frac{-1}{2}[r^2\cos^2\alpha+\frac{2a_{12}r^2\sin\alpha\cos\alpha}{\sqrt{a_{11}a_{22}}}+r^2\sin^2\alpha]\Big)\frac{r\,dr\,d\alpha}{\sqrt{a_{11}a_{22}}}$$

$$=\frac{1}{4\pi\mathcal{D}^{1/2}a_{11}a_{22}}\int_{\alpha=0}^{\frac{\pi}{2}}\sin 2\alpha\,d\alpha\int_{r=0}^{\infty}r^3\exp\Big(\frac{-r^2}{2}[1+\frac{a_{12}\sin 2\alpha}{\sqrt{a_{11}a_{22}}}]\Big)\,dr \qquad (8)$$

Next, we need to show that $\mathcal{H}:=1+\dfrac{a_{12}\sin 2\alpha}{\sqrt{a_{11}a_{22}}}\geq 0$ to ensure the expression in (8) is bounded.

First, since $\|x-y\|^2=\|x\|^2+\|y\|^2-2(x\cdot y)\geq 0 \implies \|x\|^2+\|y\|^2\geq 2(x\cdot y)$, and let the angle between the vectors $x,y$ be $\theta=\cos^{-1}\Big(\dfrac{x\cdot y}{\|x\|\|y\|}\Big)$, we have

$$\frac{\Big(\sigma_b^2+\sigma_w^2(x\cdot y)\Big)^2}{\Big(\sigma_b^2+\sigma_w^2\|x\|^2\Big)\Big(\sigma_b^2+\sigma_w^2\|y\|^2\Big)}$$

$$=\frac{\sigma_b^4+(\sigma_w^2)^2(x\cdot y)^2+2\sigma_b^2(\sigma_w^2)(x\cdot y)}{\sigma_b^4+(\sigma_w^2)^2\big(\|x\|^2\|y\|^2\big)+\sigma_b^2(\sigma_w^2)\big(\|x\|^2+\|y\|^2\big)}$$

$$=\frac{\sigma_b^4+(\sigma_w^2)^2(\|x\|\|y\|\cos\theta)^2+\sigma_b^2(\sigma_w^2)2(x\cdot y)}{\sigma_b^4+(\sigma_w^2)^2\big(\|x\|^2\|y\|^2\big)+\sigma_b^2(\sigma_w^2)\big(\|x\|^2+\|y\|^2\big)}$$

$$\leq 1.$$

This means that we can define a quantity $\phi$ as

$$\phi=\cos^{-1}\frac{\big(\sigma_b^2+\sigma_w^2(x\cdot y)\big)}{\Big((\sigma_b^2+\sigma_w^2\|x\|^2)(\sigma_b^2+\sigma_w^2\|y\|^2)\Big)^{1/2}}$$

$$=\cos^{-1}\frac{\dfrac{1}{D}\big(\sigma_b^2+\sigma_w^2(x\cdot y)\big)}{\dfrac{1}{D}\Big((\sigma_b^2+\sigma_w^2\|x\|^2)(\sigma_b^2+\sigma_w^2\|y\|^2)\Big)}$$

$$=\cos^{-1}\Big(\frac{-a_{12}}{\sqrt{a_{11}a_{22}}}\Big)$$

$$\implies \cos\phi=\Big(\frac{-a_{12}}{\sqrt{a_{11}a_{22}}}\Big) \qquad (9)$$

This also leads to

$$\mathcal{H}:=1+\frac{a_{12}\sin 2\alpha}{\sqrt{a_{11}a_{22}}}$$

$$=1+\frac{\dfrac{-1}{D}\big(\sigma_b^2+\sigma_w^2(x\cdot y)\big)\sin 2\alpha}{\dfrac{1}{D}\Big((\sigma_b^2+\sigma_w^2\|x\|^2)(\sigma_b^2+\sigma_w^2\|y\|^2)\Big)}$$

$$=1-\frac{\big(\sigma_b^2+\sigma_w^2(x\cdot y)\big)\sin 2\alpha}{\Big((\sigma_b^2+\sigma_w^2\|x\|^2)(\sigma_b^2+\sigma_w^2\|y\|^2)\Big)^{1/2}}$$

$$\geq 1-\frac{\big(\sigma_b^2+\sigma_w^2(x\cdot y)\big)}{\Big((\sigma_b^2+\sigma_w^2\|x\|^2)(\sigma_b^2+\sigma_w^2\|y\|^2)\Big)^{1/2}}$$

$$\geq 0 \qquad \blacksquare$$

With a change of variables, we now evaluate the integral involving the parameter $r$ in expression (8) as follows.

Let $\eta = \dfrac{r^2}{2}\mathcal{H}$. Then $r = \sqrt{\dfrac{2\eta}{\mathcal{H}}} \implies dr = \dfrac{1}{2}\sqrt{\dfrac{2}{\mathcal{H}}}\eta^{-1/2}\,d\eta$. We have

$$\int_{\eta=0}^{\infty}\left(\frac{2}{\mathcal{H}}\right)^{\frac{3}{2}}\eta^{\frac{3}{2}}e^{-\eta}\frac{1}{2}\sqrt{\frac{2}{\mathcal{H}}}\eta^{\frac{-1}{2}}\,d\eta$$

$$= \int_{\eta=0}^{\infty}\frac{2^2}{\mathcal{H}^2}\frac{1}{2}\eta e^{-\eta}\,d\eta$$

$$= \frac{2}{\mathcal{H}^2}\int_{\eta=0}^{\infty}\eta\,e^{-\eta}\,d\eta$$

$$= \frac{2}{\mathcal{H}^2}\Gamma(2) = \frac{2}{\mathcal{H}^2}$$

$$= \frac{2}{\left(1+\dfrac{a_{12}\sin 2\alpha}{\sqrt{a_{11}a_{22}}}\right)^2}$$

$$= \frac{2}{\left(1-\cos\phi\sin 2\alpha\right)^2}. \qquad \big(\text{from equation (9)}\big)$$

The complete expression (8) becomes

$$\frac{1}{4\pi\mathcal{D}^{1/2}a_{11}a_{22}}\int_{\alpha=0}^{\frac{\pi}{2}}\frac{2\sin 2\alpha}{\left(1-\cos\phi\sin 2\alpha\right)^2}\,d\alpha$$

$$= \frac{1}{2\pi\mathcal{D}^{1/2}a_{11}a_{22}\sin^3\phi}\Big(\sin(\phi)+(\pi-\phi)\cos(\phi)\Big). \qquad \big(\text{from equation (4)}\big)$$

where $\phi = \cos^{-1}\left(\dfrac{-a_{12}}{\sqrt{a_{11}a_{22}}}\right)$.

Finally,

$$2\pi\mathcal{D}^{1/2}a_{11}a_{22}\sin^3\phi$$

$$= 2\pi\mathcal{D}^{1/2}a_{11}a_{22}\left(1-\cos^2\phi\right)^{3/2}$$

$$= 2\pi\mathcal{D}^{1/2}a_{11}a_{22}\left(1-\frac{a_{12}^2}{a_{11}a_{22}}\right)^{3/2}$$

$$= 2\pi\mathcal{D}^{1/2}\left(a_{11}a_{22}\right)^{-1/2}\left(a_{11}a_{22}-a_{12}^2\right)^{3/2}$$

$$= 2\pi\left(\mathcal{D}^2 a_{11}a_{22}\right)^{-1/2}\left(\mathcal{D}\left(a_{11}a_{22}-a_{12}^2\right)\right)^{3/2}$$

$$= 2\pi\left(\left(\sigma_b^2+\sigma_w^2\|x\|^2\right)\left(\sigma_b^2+\sigma_w^2\|y\|^2\right)\right)^{-1/2}\left(1\right)^{3/2} \qquad \big(\text{from equation (7)}\big)$$

The expected value of the product of post-activations at the output of the $j^{th}$ hidden node in the first hidden layer is therefore determined to be

$$E[\mathbf{X}_j(x)\mathbf{X}_j(y)]$$

$$= \frac{1}{2\pi}\left(\left(\sigma_b^2+\sigma_w^2\|x\|^2\right)\left(\sigma_b^2+\sigma_w^2\|y\|^2\right)\right)^{1/2}\Big(\sin(\phi)+(\pi-\phi)\cos(\phi)\Big),$$

$$\text{where } \phi = \cos^{-1}\left\{\frac{\left(\sigma_b^2+\sigma_w^2(x\cdot y)\right)}{\left(\left(\sigma_b^2+\sigma_w^2\|x\|^2\right)\left(\sigma_b^2+\sigma_w^2\|y\|^2\right)\right)^{1/2}}\right\}.$$

To compute the expected value, $E[\mathbf{X}_j(x)] = \int \max(b + w \cdot x) f_{b_j^0, W_{jk}^0}(b, w) \, dw \, db$, we denote $U = b + w \cdot x \sim N(0, \sigma_b^2 + \sigma_w^2 \|x\|^2)$, and apply the change in variables: $\frac{1}{2\sigma^2} u^2 = t$, where $\sigma^2 = \sigma_b^2 + \sigma_w^2 \|x\|^2$ to obtain

$$
\begin{aligned}
E[\mathbf{X}_j(x)] &= \int_{-\infty}^{\infty} (u)_+ f_U(u) du \\
&= \int_0^{\infty} u \frac{1}{\sqrt{2\pi}\sigma} e^{-\frac{1}{2\sigma^2} u^2} du \\
&= \int_0^{\infty} \sigma^2 dt \frac{1}{\sqrt{2\pi}\sigma} e^{-t} \\
&= \frac{\sigma}{\sqrt{2\pi}} \\
&= \frac{\sqrt{\sigma_b^2 + \sigma_w^2 \|x\|^2}}{\sqrt{2\pi}}
\end{aligned}
$$

The covariance function at the network output is therefore determined to be

$$
E\left[ \left( b_i^1 + \sum_{j=1}^{N_1} W_{ij}^1 \mathbf{X}_j(x) \right) \left( b_i^1 + \sum_{k=1}^{N_1} W_{ik}^1 \mathbf{X}_k(y) \right) \right] - E\left[ b_i^1 + \sum_{j=1}^{N_1} W_{ij}^1 \mathbf{X}_j(x) \right] \left[ b_i^1 + \sum_{k=1}^{N_1} W_{ik}^1 \mathbf{X}_k(y) \right]
$$

$$
= E[(b_i^1)^2] + \sum_{j=1}^{N_1} E[(W_{ij}^1)^2] E[\mathbf{X}_j(x)\mathbf{X}_j(y)] - \frac{1}{2\pi} \sqrt{\sigma_b^2 + \sigma_w^2 \|x\|^2} \sqrt{\sigma_b^2 + \sigma_w^2 \|y\|^2}
$$

$$
= \sigma_b^2 + \frac{\sigma_w^2}{N_1} N_1 E[\mathbf{X}_j(x)\mathbf{X}_j(y)] - \frac{1}{2\pi} \sqrt{\sigma_b^2 + \sigma_w^2 \|x\|^2} \sqrt{\sigma_b^2 + \sigma_w^2 \|y\|^2}
$$

$$
= \sigma_b^2 + \frac{\sigma_w^2}{2\pi} \left( \sigma_b^2 + \|x\|^2 \sigma_w^2 \right)^{\frac{1}{2}} \left( \sigma_b^2 + \|y\|^2 \sigma_w^2 \right)^{\frac{1}{2}} \left( \sin \phi + (\pi - \phi) \cos \phi - 1 \right). \qquad \blacksquare
$$

