# OpenReview forum: "Guiding Neural Network Initialization via Marginal Likelihood Maximization"
_ICLR.cc/2021/Conference — Reject_

### Official Review · AnonReviewer1 · 2020-10-21
**A potentially nice idea, but more work is required**

**Rating:** 4
**Confidence:** 5

**Review:**

##########################################################################

Paper Summary:

The problem this work addresses is the design of a principled way to obtain an appropriate weight initialisation for neural networks. Building on the equivalence between a shallow, infinite width neural network and a Gaussian process with an appropriately chosen kernel function, the authors set off with the key idea of finding the optimal kernel parameters of the GP associated to a neural network, by maximising the marginal log likelihood of the GP, and use such parameters to initialise the weights of the neural network.
A simple experiment on regression over labels of MNIST demonstrates that the approach is sensible.



##########################################################################

Reasons for score:

Overall, I liked the intuition behind the idea of this work. However, I have several concerns for accepting this paper. First of all, the key to the success of current initialisation methods (e.g. He’s initialisation) lies in their simplicity and low computational costs. The proposed method, instead, requires inverting matrices, that cost $O(n^3)$. The authors acknowledge this fact in their concluding remarks, alluding at the fact that maximising the marginal log likelihood on a very small subset of training points could work, thus substantially reducing computational overheads. Unfortunately, this is not shown in the paper, and needs to be properly addressed.
Second, the analytical derivation of this work applies to shallow models, and by no means it can be applied to modern deep networks without resorting to some sort of model approximation. This is not discussed.
Third, the editorial quality of this paper can enjoy some improvements, especially concerning the mathematical notation which is inconsistent and confusing.


##########################################################################Pros:

Positive points:

1) I liked the idea of this work, and I think there is a lot of potential to develop it further. In my opinion, the authors should look at (deep) GP approximations, e.g. through random Fourier features, which are also amenable to approximately describe deep neural network architectures. The key for the success of the idea is: make sure the computational cost is commensurate to the improvements in terms of “performance metrics” with respect to existing (and much cheaper) initialisation methods.

2) The main technical contribution of this work is an alternative derivation of an appropriate covariance/kernel function that is equivalent to ReLU activation functions. The authors should focus on the key advantage of this alternative derivation. From the appendix, it seems to relate to its ability to tackle bias terms more comfortably, but from the experiments (where biases are zero) this does not seem to be sufficiently compelling.


##########################################################################

Negative points:

1) As alluded above, my main negative point relates to the computational cost of the proposed method, which is $O(n^3)$, especially when compared to existing methods that cost essentially nothing. As such, the authors should work hard in convincing the reader and the practitioner that there is really a competitive advantage in using the proposed approach. If the method was applicable to larger networks and larger datasets, its cost would be prohibitive. As such, it is important to dig more into GP model approximations, which may be the key for the applicability of the scheme in the general case (see e.g. Cutajar, ICML 17).
From a more philosophical point of view, however, I have an additional observation. Once you find the optimal parameters for the kernel function of a GP, which in the particular case of this paper is used for regression (hence, analytical solutions for the predictive distribution are available), why not using directly the GP for the problem at hand, instead of “reverting” to a simple neural network? Since the highest price is already payed (inverting a matrix), then it could be questionable to “throw away” the GP and use a neural network.

2) The notation in sec 2.1, 2.2, 2.3, and 2.4 can enjoy some polishing. There are multiple definitions that use the same symbol ($d_{in}=N_0$, $N_1=d_{in}$, $y$ is both a target variable, and a different input than $x$, $f$ is used both to indicate the input/output mapping, the GP prior on $f$, and a probability density function on the weights and biases, $U$ and $V$ have been defined as $z$ elsewhere, …)

3) The MNIST classification as regression example should be clarified. First of all, despite being a known “trick”, regressing over labels requires some care (see for example Milios, NeurIPS 18). Additionally, my understanding is that the maximisation of the marginal log likelihood is done on a “grid” of kernel parameters, instead of the more general approach using L-BFGS, gradient descent, or conjugate gradient, to name a few. This however, reiterates point 1), that is: you transform parameter initialisation of a neural network into a rather costly optimisation problem. Is it worth it?


#########################################################################

Additional comments:

I think this paper has some potential, but it requires more work: 1) to make it more practical, 2) to extend it to neural network architectures adapted to work on more “involved” data, 3) to improve exposition.

---

> ### Author Response · Authors · 2020-11-17
> **Response to Review #1**
>
> Thank you for your comments and suggestions! We appreciate this opportunity to clarify some of our findings. Our empirical results indicated that the hyperparameter estimates $(\sigma^2_w, \sigma^2_b)$ remained to be (3.6, 0.0) for all training sizes from 10000 to 50000 samples. This suggests that smaller training sets would be sufficient for marginal likelihood maximization. A key part of our current research is focusing on evaluating the actual computational time to help determine the value of our approach.
>
> We recognize that not all activation functions have corresponding closed-form covariance functions. Computing these covariance functions for deep networks would be harder without approximation. We are therefore applying the Monte Carlo approximation method in our ongoing research to validate our proposed method for other activation functions. Thank you for your comments on the mathematical notation. We are cleaning up the expressions to make them more consistent.
>
> We appreciate your suggestions on exploring deep GP approximations and your comments on the alternative derivation of covariance function. We agree that for the MNIST dataset assuming the bias term to be 0 does not affect the outcome. However, we are not certain if the same assumption holds for all datasets. Incorporating the bias term in the formulation removes the need for the assumption. In addition, the fact that our approach computes the values of post-activation instead of pre-activation allows us to formulate the recurrence relation for deep networks more easily and clearly. As pointed out in [Duvenaud et al., 2014, Cho and Saul, 2009] Mercer's theorem indicates that the covariance function can be expressed as an inner product of features. Utilizing our alternative form of covariance function and applying the Mathematical Induction we derived and proved the recurrence relation which is in a form different from Equation 11 in [Lee et al.,2018]. Unfortunately it was not added to our initial submission in time.
>
> [1]
> [RESPONSE]
> Thank you very much again for your suggestion on exploring GP model approximations. The high computational cost is indeed a main drawback of utilizing marginal likelihood maximization. Our work attempted to take advantage of the training data directly to improve neural network initialization.  Our preliminary results suggested consistent estimates of $(\sigma^2_w, \sigma^2_b)$ within the design search grid for training sizes from 10000 to 50000 samples. This indicates that smaller training sets would be sufficient for hyperparameter estimation, thus reducing computational time.  We agree that more investigation is needed to evaluate the gain in network accuracy improvement against the computation time. The work on single-hidden-layer networks and their corresponding Gaussian process models allowed us to conduct simple experiments for validating our proposed approach. Our ultimate goal is to extend the method to deep neural networks and determine if the benefit outweighs the cost for more complex tasks.
>
> [2]
> [RESPONSE]
> Thank you for pointing out the inconsistencies in notation. They all have been updated.
>
> [3]
> [RESPONSE]
> We agree that the rationale behind treating MNIST classification as regression should be clarified. Thank you also for your question on the value of our proposed method. Although our preliminary results suggest that marginal likelihood maximization may help initializing single-hidden-layer ReLU neural network on MNIST classification, the trade-off between accuracy improvement and computational time for multilayer networks is being investigated in our current research.
>
> [References]
> 1. David Duvenaud, Oren Rippel, Ryan P Adams, and Zoubin Ghahramani. Avoiding pathologies in very deep networks. AISTATS, page 9, 2014.
> 2. Youngmin Cho and Lawrence K. Saul. Kernel Methods for Deep Learning. Advances in Neural Information Processing Systems 22, pages 342{350. Curran Associates, Inc., 2009.
> 3. Jaehoon Lee, Yasaman Bahri, Roman Novak, Samuel S. Schoenholz, Jeffrey Pennington, and Jascha Sohl-Dickstein. Deep Neural Networks as Gaussian Processes. 2018 ICLR, March 2018.

---

### Official Review · AnonReviewer2 · 2020-10-27
**Promising but experimental set up is too simplistic and there is no clear improvement over simpler approaches**

**Rating:** 4
**Confidence:** 4

**Review:**

Summary:

The paper considers the initialisation of fully connected neural networks with ReLUs.
By using the GP-one hidden layer neural network equivalence result, the paper considers using the equivalent GP and its marginal likelihood to decide the initial sampling parameters for the weights and biases. The paper first re-derives the covariance function of the GP corresponding to a single-hidden-layer ReLU network. To motivate the use of the marginal likelihood, the paper provides a simulation showing optimising the marginal likelihood on the data drawn from a GP arrives at hyperparameters that closely match the hypers used to generate the data. An experiment with neural network regression on the MNIST dataset shows the initialisation based on the GP marginal likelihood is promising.

Assessment:

Strengths:

1. The submission tackles an important problem in neural networks -- initialisation as despite a suite of techniques proposed for this task, any new technique that can consistently improve over the old ones in practice or provide more theoretical insights will be beneficial to the field.

2. The topic of connecting (deep) neural networks and GPs and how to exploit this to improve neural network training is important. This submission shows this connection is useful for initialisation, as the marginal likelihood for GP regression can be computed in closed-form and for small dataset, optimising this objective is not costly.

Weaknesses:

3. novelty: The elements of the proposed approach are not novel. The covariance function for ReLU networks is not new and I don’t see the new insights from the new derivation compared to the work of Lee et al (for example: does this make things simpler when moving to deeper networks or a different activation function). The simulation study is useful, but not surprising since given enough data points the maximisers of the marginal likelihood should be the hyperparameters used to generate the data.

4. practical significance: The experiment on the MNIST dataset is promising, however, it fell short of bringing the strengths (1 and 2) home. For example, it is not clear to me why a practitioner should choose the proposed approach which requires training (potentially expensive if the subset is large) over simpler approaches like He-init --- figure 3 shows no improvement over He-init and the accuracy results in table 1 doesn’t really show a clear difference between methods at the end. I’m not entirely sure I got the reasoning of the experimental set-up, e.g. why the training sets are subsets of the full MNIST, and what is the difference in terms of results between different training regimes? The task considered here is MNIST regression with a network with only one hidden layer. It is therefore very hard to justify the ‘near-optimal’ performance claimed early in the paper. It’d be good to show that the same approach (with a new (analytic) kernel) works for different datasets, activation functions and network architectures, as promised in the future work. In practice, there are other hyperparameters for the network + training that need to be tuned, and the initialisation hyperparameters could be added to this procedure without adding much complexity.

5. clarity: Whilst the exposition of the covariance function and single hidden layer nets is great, I found the presentation for the experiments less organised. It is not clear to me what is the criterion for best and least accurate hypers and why the proposed method seems to be poorer at epoch 0 given that this initialisation is data-dependent (does this mean the marginal likelihood does not correlate with the performance at init?). It’d be clearer if some descriptions of fig 3 and table 1 (and the error bars across multiple runs) are provided.

Overall: I think the proposed method is promising and the topic is of importance. However, the submission in its current form, I think, is not ready for acceptance, primarily due to the lack of evidence of improvement over previous techniques and a more realistic experimental set-up. The additional work listed in 4 and additional experiments/clarifications will greatly strengthen the work.

Minor:

one hot encoding sentence in sec 2.1 is confusing
2.2. first paragraph and third paragraph: guassian -> gaussian


Update 1: I appreciate the authors' response. I think including the "on-going research" points as the authors brought up  and improving the clarity will greatly strengthen the submission. I still think the current form is not ready yet and would like to keep my score as is.

---

> ### Author Response · Authors · 2020-11-17
> **Response to Review #2**
>
> Thank you for your comments on the importance of network initialization and how we could strengthen our research on this topic! In particular, we appreciate you comments on the practical significance of our proposed technique [Weakness 4]. Our responses to your concerns are given below.
>
> [3.NOVELTY]
> [RESPONSE]
> Regarding the covariance function for ReLU networks, the main difference is that our approach computes the values of the post-activation rather than pre-activation. This allows us to formulate and derive more easily the recurrence relation through expressing the covariance function as an inner product in the feature space(space([Cho and Saul, 2009, Duvenaud et al., 2014]). The resulting recurrence expression (which is in a form different from Equation 11 in [Lee et al., 2018] can be straightforwardly proved using Mathematical Induction, but was unfortunately not added to our initial submission in time. Concerning the simulation study, our goal was to provide empirical evidence of the practicality of marginal likelihood maximization.  The results reassured us that this major component in our proposed method is indeed relevant.
>
> [4.PRACTICAL SIGNIFICANCE]
> [RESPONSE]
> We appreciate the simplicity and power of He-init. Nevertheless, we believe that learning from data can help neural network training. Our experiment confirmed the consistency in using marginal likelihood, implying that one only need to utilize small training sets. This will reduce the computation cost in estimating the hyperparameter pair.  Our on-going research is also assessing the benefits in accuracy improvement against the time cost.
>
> We recognize the limitations in experiments using single-hidden-layer neural networks.  Recently we conducted an experiment using a two-hidden-layer neural network. Through marginal likelihood maximization, GP-init gives the consistent estimates of the hyperparameters. Empirical results suggested a more evident performance difference and that GP-init improves NN prediction accuracy and uncertainty much faster than He-init as the neural network training size increases.
>
> We agree that there is much more work to be done. We are extending our investigation to include CIFAR-10 dataset and deeper networks. Since not all activation functions may have corresponding covariance functions in closed-form, we are experimenting with Monte Carlo approximation method.
>
> [5.CLARITY]
> [RESPONSE]
> We appreciate your comments on the experiment setting. The best and worst hyperparameter pairs were determined after the fact. Specifically, after running the experiment we recorded the hyperparameter pairs corresponding to the highest neural network prediction accuracy rate and the lowest accuracy rate. For example, for training size = 30000, the highest accuracy of 0.9750 is attained at  $(\sigma^2_w, \sigma^2_b) = (1.2, 0.0)$ while the lowest accuracy of 0.9691 is attained at  $(\sigma^2_w, \sigma^2_b) = (2.0, 2.0)$. On the other hand, He-init is fixed at $(\sigma^2_w, \sigma^2_b) = (2.0, 0.0)$, producing an accuracy rate of 0.9707. GP-init applies the recommended $(\sigma^2_w, \sigma^2_b) = (3.6, 0.0)$, producing an accuracy rate of 0.9729. The procedure was repeated for training sizes from 10000 to 50000 samples to generate Table 1, showing both the neural network prediction accuracy and the associated hyperparameter pair for each case.
>
> The objective of Figure 3 was to demonstrate that the convergence rate of GP-init method becomes more comparable to that of He-init as the training size increases. We chose He-init as the benchmark for its simplicity and effectiveness. For example, GP-init reaches He-init's level at the fourth epoch for training size = 1000, and at the third epoch for training size = 5000. We believe that GP-init will converge as fast as He-init for larger training sizes, particularly for deeper networks. This is being investigated in our ongoing research. Finally, we are unsure about the reason for unsatisfactory training accuracy at epoch 0.
>
> [MINOR]
> [RESPONSE]
> Thank you for bringing the typographical errors to our attention. They have been corrected in our revision.
> Conventionally, one-hot representation consists of all 0's except at the location (to be designated as a 1) associated with the digit it is representing. Following [Lee et al., 2018], we modified the one-hot representation, replacing 0 with -0.1, and 1 with 0.9, to ensure it has mean = 0 and variance = 0.1.
>
> [References]
> 1. David Duvenaud, Oren Rippel, Ryan P Adams, and Zoubin Ghahramani. Avoiding pathologies in very deep networks. AISTATS, page 9, 2014.
> 2. Youngmin Cho and Lawrence K. Saul. Kernel Methods for Deep Learning. Advances in Neural Information Processing Systems 22, pages 342{350. Curran Associates, Inc., 2009.
> 3. Jaehoon Lee, Yasaman Bahri, Roman Novak, Samuel S. Schoenholz, Jeffrey Pennington, and Jascha Sohl-Dickstein. Deep Neural Networks as Gaussian Processes. 2018 ICLR, March 2018.

---

### Official Review · AnonReviewer4 · 2020-10-28
**Raises an interesting question. Unfortunately, doesn't answer it.**

**Rating:** 4
**Confidence:** 4

**Review:**

### UPDATE:

Per my response in the thread, I appreciate the authors' replies and updates, but I am keeping my score because
- Even with the new 2-layer results, I find this experimental setting still too limited;
- Even within the conducted experimental setting, the benefit of likelihood-guided initialization over He init is not robust (notably on small dataset sizes, which is contrary to the expectation that a good prior will be more beneficial in such cases).

### SUMMARY:
The paper proposes to use the marginal likelihood of the Gaussian process (GP) corresponding to the infinite width Bayesian neural network (NN) to select hyper-parameters of the finite width, SGD-trained NN (instead of evaluating performance on the validation set).

Experimentally, weight and bias variance in 1-hidden layer ReLU network are considered on MNIST, and on multiple training set sizes GP-likelihood-selected configuration slightly outperforms the default He-initialization, and slightly under-performs the best possible configurations.

The paper also provides an alternative derivation of the ReLU kernel.


### REVIEW SUMMARY:
While I appreciate and support the question (**Q: Can we use NN-GP likelihood to select hyper-parameters of a finite SGD-trained NN?**) the paper raises, I believe the paper does not answer it (due to very limited experimental evidence), and raising the question itself is not sufficient for publication.


### PROS:
1. I believe the proposed idea is promising.
2. Answering the question **Q** is certainly of interest to the research community. Unlike training and evaluating on a validation set, marginal likelihood of the NN-GP can be simply evaluated in closed form on a small subset of the training set, potentially making it a cheap proxy for validation performance.
2. The paper is clearly written and easy to follow.


### CONS:
1. My key concern: unfortunately experimental validation of the hypothesis is not adequate. After reading the paper I remain in the dark regarding whether I should use NN-GP likelihood for hyper-parameter search or not. Precisely:

	1. The idea is only evaluated on a single hidden layer, fully connected network, on MNIST (Table 1).
    	1. MNIST is a very simple dataset, and a fully connected network is not well-suited for image classification.
		2. The differences between test performance of best and worst configurations are within 1%, and differences between He-Init and NN-GP-guided configs are within 0.25%.
		3. Notably, the likelihood-guided config performs worse than He-Init on the smallest training set size (10000). This is counter to the hypothesis that NN-GP likelihood evaluates the suitability of the NN prior, since we expect the quality of the prior to become more and more important as the training set becomes smaller.
		4. Finally, the likelihood-guided config of (3.6, 0) is at the corner of the hyper-parameter grid, which makes it unclear what would be observed if the grid of weight variances was extended further.

    Due to all of the above, I find this experiment unconvincing. The hypothesis should be evaluated on a hyper-parameter grid that spans a large range of model performances (vs fractions of a percent), one where likelihood is not maximized at an edge of the grid, as well as considering other datasets (e.g. CIFAR-10), other hyper-parameters (e.g. depth), and more appropriate architectures (e.g. CNNs for image classification tasks). Note that there are a many other nuances that would need to be addressed to properly answer the the question **Q** (e.g. comparing computational expense of NN-GP-based search to those of grid/random/Bayesian search, including tuning the additional meta-hyper-parameters (NN-GP training set size, diagonal regularization), discussion of the additional technical complexity and feasibility of evaluating the NN-GP on practical architectures, how wide the finite network has to be in practice etc). It would be OK to not touch on these in the same paper, and only provide a convincing answer to a more specific question like **"Q0: How well does NN-GP likelihood correlate with SGD-NN generalization?”**. Unfortunately, the paper doesn’t answer this either.

	2. The idea itself is fairly straightforward, not particularly motivated theoretically (e.g. is there a bound on the difference in performance between NN-GP and SGD-trained NNs, perhaps in some toy cases?), and arguably not novel, as it is raised in one of the referenced papers ([Deep Neural Networks as Gaussian Processes, Lee et al 2018](https://arxiv.org/abs/1711.00165), page 6, second to last paragraph before section 3). This is to highlight that I see most of the potential value of this paper in answering the question **Q/Q0** (vs raising it), hence putting so much importance on my prior point. This would not have been an issue for me if the question was answered.

### QUESTIONS:
1. Could you please provide data (if it is present) for Table 1 for training sizes of 1000, 3000, 5000? These appear in Figure 3, but not Table 1.
2. To tackle **Q0**, I encourage the authors to plot NN-GP likelihood vs test performance for all configurations that were considered (in addition to Table 1). To be clear, I’m afraid this would still not be sufficient to make a convincing case due to the limitations of the considered grid search discussed above.
3. I am not certain/convinced with the point Figure 3 is making: is it that training curves of He-init and NN-GP-guided become more similar with increasing training set size? If so, I find the effect to be too small to be convincing on Figure 3 (perhaps zooming in  / measuring numerically could help), but I am also not sure why this would be an interesting/important metric to look at (vs test accuracy, as measured in Table 1).

---

> ### Author Response · Authors · 2020-11-17
> **Response to Review #4**
>
> Thank you for your comments and suggestions! Our intention was to show that one can benefit from utilizing data to find appropriate initial hyperparameter setting and that Gaussian process models provide such tool, as mentioned in [Lee et al., 2018]. We appreciate your comments on using CIFAR-10 dataset, an expanded search grid, and convolutional rather than fully-connected neural network models. These are all currently being studied in our ongoing research.  We also appreciate your suggestions on investigating a bound on the performance difference between NN-GP and SGD-trained NNs.
>
> To supplement our previous results, we recently conducted an experiment using a two-hidden-layer neural network. The new results helped to strengthen our case, albeit the model is still shallow. Boxplots generated from these results indicated that GP-init improves NN prediction accuracy and uncertainty much faster than He-init as the training size increases. We speculate that stronger evidence may be possible for deeper networks. Part of the new results are as follows:
>
> Mean prediction accuracy
> Training size: (1000, 3000, 5000, 7000, 9000)
> "Best" case(after the fact result): 0.9290, 0.9527, 0.9644, 0.9689, 0.9712
> "Worst" case(after the fact result): 0.8594, 0.9295, 0.9296, 0.9363, 0.9400
> He-init: 0.9259, 0.9520, 0.9626, 0.9653, 0.9657
> GP-init: 0.9105, 0.9504, 0.9639, 0.9685, 0.9711
>
> Standard deviation (x $10^{-4}$)
> Training size: (1000, 3000, 5000, 7000, 9000)
> "Best" case(after the fact result): 8.5, 12.6, 7.6, 11.6, 8.6
> "Worst" case(after the fact result): 95.4, 31.3, 29.9, 26.2, 16.2
> He-init: 20.1, 14.1, 8.8, 12.3, 21.4
> GP-init: 36, 18.3, 12.4, 14.1, 10.4
>
>
> [RESPONSE 1]
> Prediction accuracy
> Training size: (1000, 3000, 5000)
> "Best" case(after the fact result): 0.9268, 0.9507, 0.9553
> "Worst" case(after the fact result): 0.8818, 0.9402, 0.9431
> He-init: 0.9212, 0.9459, 0.9531
> GP-init: 0.9070, 0.9482, 0.9541
>
> Hyperparameter pair
> Training size: (1000, 3000, 5000)
> "Best" case(after the fact result): consistent: (0.4, 0.0)
> "Worst" case(after the fact result): (3.6, 2.0), (0.4, 2.0), (0.4, 2.0)
> He-init: fixed at (2.0, 0.0)
> GP-init: consistent recommendation: (3.6, 0.0)
>
>
> [RESPONSE 2]
> Without a set of generally acceptable guiding principles, we found it challenging to decide on the search ranges for the hyperparameters. However, recent research works gave us a rough idea of what they could be.  In particular, Schoenholz et al. [2017] choose $1 \le \sigma^2_w \le 4$ and $\sigma^2_b = 0.05$ for their experiments on MNIST and CIFAR-10. In addition, Lee et al. [2018] design their search grid with $\sigma^2_w$ ranges from 0.1 to 5.0, and $\sigma^2_b$ ranges from 0 to 2.0 for their experiments on MNIST and CIFAR-10 datasets.  Considering these options and the capability limitations in our available hardware, we chose a search grid with $\sigma^2_w \in (0.4, 1.2, 2.0, 2.8, 3.6)$ and $\sigma^2_b \in (0.0, 1.0, 2.0)$. We intend to use a much finer grid in our future study to investigate the behavior of the models more precisely.
>
>
> [RESPONSE 3]
> We recognize that one advantage of using He-initialization scheme is its rapid convergence to solution [He et al., 2015]. The main purpose for generating Figure 3 was to show that the convergence rate of GP-init method becomes more comparable to that of He-init as the training size increases. For example, GP-init reaches He-init's level at the fourth epoch for training size = 1000, and at the third epoch for training size = 5000. We suspect that GP-init will converge as fast as He-init for larger training sizes. We are currently investigating model convergence for deeper neural networks
>
>
> [References]
> 1. Kaiming He, Xiangyu Zhang, Shaoqing Ren, and Jian Sun. Delving Deep into Rectifiers: Surpassing Human-Level Performance on ImageNet Classification. 2015 IEEE Conference on Computer Vision and Pattern Recognition (CVPR), February 2015.
> 2. Jaehoon Lee, Yasaman Bahri, Roman Novak, Samuel S. Schoenholz, Jeffrey Pennington, and Jascha Sohl-Dickstein. Deep Neural Networks as Gaussian Processes. 2018 ICLR, March 2018.
> 3. Samuel S. Schoenholz, Justin Gilmer, Surya Ganguli, and Jascha Sohl-Dickstein. Deep Information Propagation. ICLR 2017, April 2017.

---

> > ### Comment · AnonReviewer4 · 2020-11-22
> > **Thank you for the detailed reply, but experimental validation is still very limited**
> >
> > Thank you for your replies and updated experimental results. Unfortunately I am inclined to maintain my rating since:
> > 1. Even with the new 2-layer results, I find this experimental setting still too limited;
> > 2. Even within the conducted experimental setting, the benefit of likelihood-guided initialization over He init is not robust (notably on small dataset sizes, which is contrary to the expectation that a good prior will be more beneficial in such cases).
> >
> > Best,
> > R4.

---

### Official Review · AnonReviewer3 · 2020-10-28
**Finding initialization rules through the study of Gaussian Processes: one-layer case**

**Rating:** 3
**Confidence:** 3

**Review:**

### Summary
The authors propose a rule for neural network (NN) initialization, which takes into account input data.

They suppose that weights and biases of a NN are randomly drawn resp. from $\mathcal{N}(0, \sigma_w^2 / N)$ and $\mathcal{N}(0, \sigma_b^2)$, where $N$ is the number of inputs. Then, they are able to compute *explicitly* the covariance matrix of the corresponding Gaussian Process. Since an explicit result is needed, they concentrate on one-layer NNs.

They use their explicit formula for the covariance matrix to compute the likelihood of the data, given $\sigma_w^2$ and $\sigma_b^2$. Therefore, they are able to select the best pair $(\sigma_w^2, \sigma_b^2)$ according to data, i.e., maximizing the likelihood.

### Clarity
I did not understand the experimental setup presented in Section 2.5. For instance, the train/test location are supposed to lie in the interval $[0, 1]$, but the test location points lie in $[0, 2]$ in the graphs.
Besides, all the computation of the covariance should be put in appendix.

### Significance
Since the paper relies on an explicit computation in one-layer NNs, the presented method has a very low significance. At least, the authors should propose an application in deeper NNs, even by making strong approximations. As such, no clue about any generalization is provided.

Edit:
### Rebuttal
I did read the authors' rebuttal, and the main issue, i.e. the significance, has not been addressed. I cannot take into account the new experiments, since they are not in the paper. Anyway, an experiment with a 2-layer network would be a significant modification of the present paper, which would be not acceptable during the rebuttal phase.

---

> ### Author Response · Authors · 2020-11-17
> **Response to Review #3**
>
> Thank you for your comments and suggestions! Specifically we appreciate your comments on the need for generalization of our proposed method. The preliminary results presented in this submission led us to believe that the connection between neural networks and Gaussian processes can indeed be leveraged to help network training. To continue our on-going research, we have been focusing on investigating multi-layer neural networks and corresponding Gaussian process models to validate the usefulness of marginal likelihood maximization technique.
>
> [Clarity]
> I did not understand the experimental setup presented in Section 2.5. For instance, the train/test location are supposed to lie in the interval [0, 1], but the test location points lie in [0, 2] in the graphs. Besides, all the computation of the covariance should be put in appendix.
>
> [RESPONSE]
> Thank you for bringing this error to our attention. It has been corrected in our revision.
>
> The reason for showing a brief description of the computation of covariance function in Section 2.4 was that we would like to demonstrate an approach alternative to that provided in [Cho and Saul, 2009], and that the detailed derivation would be shown in the Appendix.
>
>
> [Significance]
> Since the paper relies on an explicit computation in one-layer NNs, the presented method has a very low significance. At least, the authors should propose an application in deeper NNs, even by making strong approximations. As such, no clue about any generalization is provided.
>
> [RESPONSE 2]
> We recognize the limitations in a neural network with single hidden layer. The simplicity of the structure, on the other hand, allowed us to conduct simple experiments for proof of concept. To improve confidence in our proposed approach, we recently conducted an experiment using a two-hidden-layer neural network on MNIST dataset. We measured both model prediction accuracy and uncertainty. The new empirical results indicated that GP-init improves NN prediction accuracy and uncertainty much faster than He-init as the training size increases. Our hypothesis is that experiments using deeper neural networks will provide evidence that GP-init technique is a robust alternative to He-init method.
>
> 1. Youngmin Cho and Lawrence K. Saul. Kernel Methods for Deep Learning. Advances in Neural Information Processing Systems 22, 2009.

---

### Decision · Program_Chairs · 2021-01-07
**Final Decision**

**Decision:**

Reject

**Comment:**

Addressing the initialization issue in DNNs is an important topic, and the proposed approach is found by the reviewers to be interesting. However, the reviewers feel that to clearly promote this research beyond the 'proof of concept' phase, deeper investigation in multi-layer architectures would be required. This would raise the significance of the paper. Besides extending the study to deeper networks, the paper could also benefit from elaborate experiments to increase convincingness, in particular by addressing R4's concerns regarding robustness of performance e.g. on small dataset sizes. Finally, the methodology is sound and the authors clarify the significance of the ReLU associated covariance; however, overall the paper does not offer significant technical advancements that could make up for the shortcomings in the areas discussed above.